# Applying MMD Data Mining to Match Network Traffic for Stepping-Stone Intrusion Detection

**DOI:** 10.3390/s21227464

**Published:** 2021-11-10

**Authors:** Jianhua Yang, Lixin Wang

**Affiliations:** TSYS School of Computer Science, Columbus State University, Columbus, GA 31907, USA; wang_lixin@ColumbusState.edu

**Keywords:** stepping-stone, intrusion detection, network security, MMD, data mining, packet-matching

## Abstract

A long interactive TCP connection chain has been widely used by attackers to launch their attacks and thus avoid detection. The longer a connection chain, the higher the probability the chain is exploited by attackers. Round-trip Time (RTT) can represent the length of a connection chain. In order to obtain the RTTs from the sniffed Send and Echo packets in a connection chain, matching the Sends and Echoes is required. In this paper, we first model a network traffic as the collection of RTTs and present the rationale of using the RTTs of a connection chain to represent the length of the chain. Second, we propose applying MMD data mining algorithm to match TCP Send and Echo packets collected from a connection. We found that the MMD data mining packet-matching algorithm outperforms all the existing packet-matching algorithms in terms of packet-matching rate including sequence number-based algorithm, Yang’s approach, Step-function, Packet-matching conservative algorithm and packet-matching greedy algorithm. The experimental results from our local area networks showed that the packet-matching accuracy of the MMD algorithm is 100%. The average packet-matching rate of the MMD algorithm obtained from the experiments conducted under the Internet context can reach around 94%. The MMD data mining packet-matching algorithm can fix the issue of low packet-matching rate faced by all the existing packet-matching algorithms including the state-of-the-art algorithm. It is applicable to network-based stepping-stone intrusion detection.

## 1. Introduction

Most professional hackers launch their attacks not for fun, but for gaining benefits, such as money and/or valuable information. In order to avoid detection, they normally access a remote server via a long connection chain, which spans multiple computing hosts distributed all over the world. The hosts used in a connection chain are called stepping-stones [1]. In this section, we will introduce stepping-stone intrusion, different approaches to detection stepping-stone intrusion, as well as our contributions.

### 1.1. Stepping-Stone Intrusion

A stepping-stone host can be any computers on the Internet with a relayed pair of an incoming and an outgoing connection. The concept of a stepping-stone was first introduced in the paper “Hold Intruders Accountable on the Internet” published in 1995 [2]. However, the detection of stepping-stone intrusion was not extensively discussed until 2000; since then, stepping-stone has been widely used by intruders to launch their attacks through the Internet. A stepping-stone can be used singularly, as in the case that an attacker connects into a host used as a stepping-stone and then connects out to a victim host. This way, chaining three computer hosts together is not frequently used by attackers since it is trivial to trace hackers back. One popular way to use stepping-stones is to establish a long connection chain spanning multiple hosts. The attacks behind a long connection chain are hard to be detected and tracked. Most intruders using a long connection chain to launch their attacks can be well protected. The primary reason is that it is difficult to trace back the intruders via a long connection chain due to political and geographical reasons. The attacks launched via one or more stepping-stones are called stepping-stone intrusions (SSI). The approaches developed to detect such attacks are called stepping-stone intrusion detection (SSID). SSID contains the techniques of host-based stepping-stone intrusion detection (HSSID) and network-based stepping-stone intrusion detection (NSSID).

### 1.2. Host-Based Stepping-Stone Intrusion Detection

If a host is used as stepping-stone, the host must have a relayed pair of incoming and outgoing connections. The approaches developed to detect stepping-stone intrusion based on the comparison between the incoming and outgoing connections of a host are called host-based stepping-stone intrusion detection. The key idea for HSSID is to decide if a host is used as a stepping-stone. On the other hand, if a host is used as a stepping-stone, it does not guarantee an attack exists. HSSID techniques may bring false positive errors because many legal applications use stepping-stones. To determine if a host is used as a stepping-stone, we need to decide if there is a relayed connection pair passing through the host. After examining all the incoming and outgoing connections of a host, if there exists a relayed connection pair, we can tell that the host is used as a stepping-stone.

There have been many techniques developed to examine if a host is used as a stepping-stone since 1995. Comparing the packet contents from an incoming connection with an outgoing one of a host is an obvious way to determine if a host is used as a stepping-stone. Another approach is to compare packets’ timestamp gaps. If we can count the number of packets from both the incoming and outgoing connections of a host, it is easy to determine if a host is used as a stepping-stone by simply comparing the two numbers. Other approaches developed to detect stepping-stone include random-walk and crossover packets. In the section of literature review, we will briefly introduce the most popular HSSID approaches developed to detect stepping-stone intrusion including content thumbprint, time thumbprint, packet count, random-walk and crossover packet.

### 1.3. Network-Based Stepping-Stone Intrusion Detection

HSSID techniques come with some limitations. The most critical one is that it may bring false positive errors if determining an attack completely depends on whether a host is used as a stepping-stone. The fact is that some legitimate applications may use computers as stepping-stones. One example is web accessing on the Internet. Accessing a web server via a browser may result in accessing a database server through the web server because most web forms need data from its corresponding database, which normally resides in another separate remote server. In this scenario, from the browser to the web server, then to the database server, the web server obviously plays a role of a stepping-stone. However, this is definitely not a stepping-stone intrusion.

To overcome the limitations of HSSID techniques, some other algorithms to detect SSI via estimating the length of a connection chain have been developed since 2002. These types of approaches to detect SSI are called network-based stepping-stone intrusion detection. The longer a connection chain, the higher suspicious the user uses the chain to launch attacks. It makes a lot more sense to detect SSI attacks based on the number of hosts used. Detecting an intrusion using the length of a connection chain may introduce false negative detection errors. The reason is that we can only estimate the length of the downstream section in a connection chain, other than the whole connection chain. A stepping-stone host running a detection algorithm is called a sensor. If a sensor happens to be close to the victim host in a long connection chain, the length of the downstream connection chain would be trivial. In such a case, the upstream connection dominates the length of the whole connection chain. Currently, it is still an open problem to estimate the length of an upstream connection. If a downstream connection has more than three hosts used as stepping-stones, plus the length of the upstream connection, it would definitely be more than three hosts compromised.

### 1.4. Significance of Packet-Matching to SSID

After examining the most HSSID and NSSID techniques developed so far to detect stepping-stone intrusions, we found that most of them need matching TCP/IP packets. The methods of exploiting packet-matching to detect a stepping-stone intrusion can overcome some issues faced by other detection algorithms, such as weakness to resist intruders’ session manipulation evasion and high false positive errors. Packet-matching can bring down SSID false positive errors. We understand that simply inferring an intrusion based on if a host is used as a stepping-stone is the primary reason to incur false positive detection errors. An investigation shows that even though there are some applications taking advantage of stepping-stones legitimately, the cases using three or more hosts as stepping-stones are rarely observed. One reason is that the more stepping-stones used to access a server, the less efficient of the network communication. If a server can be accessed directly, there is no any reason to be accessed indirectly via three or more hosts. We assume that it is reasonable to infer an intrusion if three or more hosts are used as stepping-stones.

Packet-matching can also resist intruders’ evasion, especially the time-jittering and chaff-perturbation manipulation. If we treat a regular network traffic as a signal, time-jittering and chaff-perturbation can be considered to a noisy signal. The noisy signal can affect the performance of an intrusion detection, but they can be filtered out by packet-matching. Time-jittering can hold a request for a while and then release it to the Internet. Chaffed packets can be easily filtered out by packet-matching because each injected meaningless packet must be removed before it arrives at the destination. Otherwise, it would affect the command execution. Therefore, chaffed packets do not have any matched packets.

### 1.5. Contribution and Novelty

Packet-matching in an interactive TCP session is to find a response packet for a request packet. The response packet is called an Echo packet and the request packet is called a Send packet. The methods developed to match TCP Send and Echo packets include using packets’ sequence number, First-Match packet-matching algorithm [3], Packet-matching Conservative algorithm [4] and Packet-matching Greedy Algorithm [4]. These algorithms all suffer from a low packet-matching rate.

In this paper, we first distinguish a Send from an Echo packet in a TCP interactive session, then model a network traffic as the RTTs of a connection chain. We second present the rationale to use the RTTs of a connection chain to represent the length of the chain. In order to obtain the RTTs of the sniffed packets from a connection, packet-matching algorithm is required. Third, we propose a novel packet-matching algorithm to match the network traffic collected from a connection chain: MMD data mining algorithm. This algorithm to match TCP packets exploits the RTTs distribution: Poisson distribution [5,6]. The idea using MMD data mining to match packets is that the RTTs from the matched packet pairs can be mined into the same cluster. MMD data mining algorithm was applied to estimate the length of a connection chain to detect a stepping-stone intrusion [6,7]. However, to the best of our knowledge, it is the first time we propose to apply MMD data mining algorithm to match network traffic.

The experimental result is encouraging. It showed that MMD data mining packet-matching algorithm outperforms all the existing packet-matching algorithms including the state-of-the-art packet-matching algorithm. It fixes the issue of a low packet-matching rate faced by all the existing packet-matching algorithms. The experimental results have shown that the packet-matching rate of the MMD data mining algorithm under the Internet context can reach around 94%. This is a very encouraged result by comparing with all the existing packet-matching algorithms.

### 1.6. Layout of the Paper

The rest of the paper is laid out as the following. Section 2 will review the related literatures. Section 3 describes the idea of modeling network traffic. Section 4 presents the rationale for packet-matching. Section 5 shows the details of MMD data mining to match network traffic. Section 6 gives the experimental packet-matching results and its analysis. We conclude the whole paper in Section 7 and discuss the future work.

## 2. Literature Review

Since 1995, many algorithms for stepping-stone intrusion detection have been proposed and some of them are closely related to the work we conducted in this paper. A typical approach to detect stepping-stone intrusion is to compare the incoming connections with the outgoing ones of the same host. If a matched pair of two connections of a host is identified, then the host is most likely used as a stepping-stone for intrusion. Another effective approach for stepping-stone intrusion detection is to estimate the length of a connection chain. The former one is called HSSID, the latter one is called NSSID.

Staniford-Chen et al. [7] first discovered a “content-thumbprint method” in stepping-stone detection. The approach was based on the content of unencrypted packets. A thumbprint is a short summary of a captured packet and used in [7] to determine whether the bidirectional packets have the exact same content. If so, the sensor host is used as a stepping-stone. A drawback of this approach is that it only works for unencrypted network traffic like a connection created using Telnet, or rlogin. If a connection chain was established using OpenSSH, then the network traffic is encrypted and this algorithm could not work.

In order to overcome the primary issue of the content-thumbprint, Zhang et al. [1] proposed a “time-thumbprint” method using timing correlation of different connections to detect stepping-stone intrusions. Time-thumbprint makes use of the timestamp of each packet to decide whether there is a relayed connection pair. This approach works for encrypted network traffic to detect stepping-stone intrusion because the packet timestamps are usually not encrypted. This algorithm uses reduced workload by only keeping the headers of the captured packet and can output good accuracy. This type of approach to detect stepping-stone intrusion can be easily defeated by session manipulation, such as chaff-perturbation and time-jittering.

In order to effectively identify stepping-stone intrusions in the presence of intruders’ session manipulation, He et al. proposed the packet counting method in [8]. In the paper, they proposed two activity-based algorithms to detect stepping-stone intrusion by taking into consideration stepping-stone connections are subject to packet-conserving transformations by attackers. Unfortunately, this method can only tolerate a small percentage of chaff rate.

Wang et al. [9] developed an interesting framework called “sleepy watermark tracing” for stepping-stone intrusion detection. This algorithm is “sleepy” because overhead will not be produced when no malicious intrusion is found. In addition, there will be some overhead when an intrusion is detected. In case of an intrusion, the victim machine then injects a watermark into the backward connection to the attacker host that initiated the intrusion and then collect information from the connection chain through collaborating with routers. The big limitation for this approach is that inserting watermark to network traffic may incur large computations and, thus, lower the efficiency of network communication. Sometime, chaff-perturbation may breakdown a watermark inserted.

There were several NSSID approaches that have been proposed to estimate the length of a connection chain. The first approach of estimating the length of a connection chain was proposed by Yung [10]. The key idea used in the paper was to estimate the length of a connection chain by calculating a Send packet’s RTT and matching it with an ACK sent by the next adjacent host. This detection approach suffered from generating false negative error as this method used the ACK packets instead of the echo packets sent from the victim host. With the approach to set up the connection chain in [10], it is impossible to capture the actual echo packets. Yung’s approach suffers from the Yardstick problem [2], as well as the issue of inaccurate packet matching.

Yang et al. [2] improved the algorithm proposed in [10] and developed the step-function detection approach to estimate the length of a connection chain. The step-function method significantly brought down both the false positive and negative errors, compared to Yung’s method proposed in [10]. In [2], a connection chain created initially contains only one connection. Then, the connection chain can be extended to two connections, three connections and even more connections. The key idea of the step-function method is to obtain the RTTs by matching a Send packet with its actual corresponding Echo packets. However, this method only worked effectively within a LAN. On the Internet, the packet-matching rate from the step-function approach is low and the packet matching accuracy is a concern.

The clustering-partitioning data mining approach proposed in [5] by J. Yang et al. is the best-known existing method among this type of detection approaches. This detection algorithm is a global approach by which it checks all the packets together to make sure that the correct RTTs are among these time differences, improving the previous known approaches to obtain the length of a connection chain. This approach works inefficient if applying to a large amount of packet set.

Yang and Lee [11] developed the TCP/IP packets cross-matching approach to detect stepping-stone intrusion and resist intruders’ evasion. Zhang et al. [12] proposed resisting intruders’ manipulation using context-based TCP/IP packet-matching. Dynamic programing and data mining techniques were used to detect stepping-stone intrusion [13,14]. Sheng et al. [15] proposed to detect stepping-stone intrusion via mining network traffic. Yang et al. [16] developed an algorithm to detect stepping-stone intrusion and resist intruders’ chaff attack based on packet cross-matching and RTT-based random walk. All the above approaches [11,12,13,14,15] suffer from a low packet-matching rate and, thus, cannot resist intruders’ chaff-perturbation rate to a high percentage.

In [17], the authors proposed a novel framework to simulate stepping-stone operations in real world that includes taking advantage of effective evasion tools and releasing a large dataset, which were used to evaluate the detection rates for stepping-stone intrusions for eight existing state-of-the-art algorithms in the literature. This is not a method developed to detect stepping-stone intrusion. On the contrary, it is a framework to simulate a stepping-stone intrusion, but it can help researchers to build a platform to study the characteristics of stepping-stone intrusion.

A tool to sniff and chaff an interactive TCP session was developed in 2018 [18]. Using this tool, researchers on stepping-stone intrusion detection can easily test if an algorithm developed can resist intruders’ manipulation. This is a tool developed to help researchers to manipulate a TCP session. Wang et al. [19] developed a new algorithm by applying k-means clustering algorithm to network traffic to detect stepping-stone intrusion. They found the algorithm can resist intruders’ chaff-perturbation evasion to a certain degree, but not a high percentage. Most research projects focused on designing algorithms to detect stepping-stone intrusion and resist intruders’ chaff evasion. However, few works on resisting intruders’ time-jittering manipulation. Wang et al. [20] proposed a framework to test the resistance of the detection algorithms on time-jittering evasion. From the above, we have seen that most algorithms developed to detect stepping-stone intrusion need an approach to match TCP packets with a high matching rate.

Wang et al. [21] proposed an efficient algorithm to remove most of the RTT outliers of the captured packets in the context of the Internet and then developed an efficient SSI detection method by using an improved version of k-Means clustering. However, this paper did not provide any analysis whether the proposed detection method for SSI is resistant to hackers’ session manipulation with either time-jittering or chaff-perturbation.

Gamarra et al. [22] proposed a model that describe the characters of stepping-stone attacks in the Internet of Things (IoT) by constructing a graph based on vulnerabilities. In this paper, the stepping-stone dynamic is formulated as a linear system. The proposed framework is further expanded to a more realistic case when the vulnerability graph evolves because the intrusion has been detected by the preinstalled intrusion detection system. A limitation of this paper is that stepping-stone intrusion with hackers’ evasion manipulations is not considered and analyzed.

Alghushairy et al. [23] proposed a new outlier detection method that can be used for network intrusion detection. The proposed Genetic-based Incremental Local Outlier Factor is designed to address the issues of existing outlier detection algorithms. The algorithm works well without any previous knowledge of data distribution and it executes in a limited memory. However, the accuracy and efficiency of the proposed outlier detection algorithm in data stream processing and mining can still be improved.

## 3. Modeling Network Traffic

The goal to model network traffic is to discover a novel algorithm to detect a stepping-stone intrusion. One popular way to detect a stepping-stone intrusion is to estimate the length of a connection chain since it is highly suspicious if a connection chain containing three or more connections is used to access a remote server. In order to estimate the length of a connection chain used in computer network communication, we model network traffic as the collection of Send and Echo packets, as well as the collection of RTTs. Computer network communication involves different types of network protocols. Network traffic may contain different types of packets in different applications. In this project, we focus on studying an interactive TCP session built using OpenSSH. In a TCP session, we can sniff Send and Echo packets, which will be defined formally later on. RTTs can be obtained from the matched Send and Echo packets sniffed from a TCP connection.

### 3.1. Connection Chain Definition

Stepping-stone intruders can make a connection chain as shown in Figure 1 using OpenSSH to launch their attacks. In Figure 1, we assume Host 0 is used by an intruder to launch an attack to Host *N* via compromised hosts Host 1, Host 2, …, Host *i* − 1, Host *i*, Host *i* + 1 and Host *N* − 1, which are stepping-stones. Stepping-stone intrusion detection can occur at one of the stepping-stones. It is assumed that the detection program resides in Host *i*. As discussed before, we call this host a detection sensor or a sensor. Stepping-stone intrusion detection is to determine if Host *i* is used as a stepping-stone. The connection from Host *i* − 1 to Host *i* is called an incoming connection of Host *i* and the connection from Host *i* to Host *i* + 1 is called an outgoing connection of the same host. The condition that Host *i* is used as a stepping-stone is that there is at least one relayed session pair between all the incoming connections and all the outgoing connections of Host *i*. This connection chain contains *N* connections including the ones Host 0 to Host 1, Host 1 to Host 2, …, Host *i* − 1 to Host *i*, Host *i* to Host *i* + 1, … and Host *N* − 1 to Host *N*. Each request packet sent from Host 0 arrives at Host 1 first, then to Host 2 from the connection Host 1 to Host 2 and, finally, come to the last one Host *N* that is the end of the connection chain. Host *N* processes each request packet received and sends a response packet back to Host 0 along the connection chain reversely. To the sensor Host *i*, the connection chain from Host *i* to Host 0 is called an upstream connection and the one form Host *i* to Host *N* is called a downstream connection.

### 3.2. Define Send and Echo Packets

As we mentioned in Section 3.1, each request packet is generated at the first host of the connection chain and forwarded along the connection chain until to the last host of the chain. The host at the end of the connection chain replies a request received and the response can go back to the first host along the connection chain in a reverse direction. Each request forwarded from Host *i*, obviously, can be acknowledged by its adjacent host Host *i* + 1 first, then echoed by a response forwarded from Host *i* + 1 to Host *i*. Sometimes, an acknowledgement and a response packet for the same request can be combined together in a TCP packet with flags Push and Ack set up. The request and response packets are critical to estimate the length of a connection chain since they travel across the whole connection chain. In the convenience of describing packet-matching algorithm, we define a Send as a request packet and an Echo as a response packet. For a specific sensor host, such as Host *i* with IP address ip-*i*, in a connection chain, a Send of Host *i* is defined as a packet collected at Host i with source IP: ip-*i*, destination port number: 22 and Push flag set up. Similarly, an Echo of Host *i* is defined as a packet collected at Host *i* with destination IP: ip-*i*, source port: 22 and Push flag set up.

As shown in Figure 2, three packets were collected from a host with IP address: 192.168.1.112 in a connection chain established from a host in Columbus State University, GA to an AWS server in Amazon Cloud. The first packet was sent from a host 192.168.1.112 to the AWS host running an OpenSSH server with IP 3.142.156.175 and port number 22. Its flags [P.] indicate that this is a packet with combined Push and Ack flags set up. So this is a Send from host 192.168.1.112. The second packet is an Echo of the host 192.168.1.112 received from the OpenSSH server with its Push and Ack set up. The third packet is an acknowledgement to acknowledge the Echo received from the server. Its flags [.] indicate that this is an acknowledgement packet only. It does not carry any data since its packet length is 0. In the following sections, we denote a Send as “S”, an Echo as “E” and an Acknowledgement as “A”. If we have *n* Send packets collected, they can be denoted as a packet queue {S_1_, S_2_, S_3_, …, S*_n_*} based on the order of their timestamps; *m* Echo packets can be denoted as {E_1_, E_2_, E_3_, …, E*_m_*}; and, similarly, *k* Ack packets denoted as {A_1_, A_2_, A_3_, …, A*_k_*}.

### 3.3. RTT from a Matched Packet Pair

A matched packet pair contains a request packet and its corresponding response packet. A matched Send and Echo pair is that the Echo must be an echoed packet of the Send collected from the same host in the same connection chain. In most cases, one Send is echoed by one Echo. In Figure 2, it shows such a case, which is easy to match a Send with an Echo. A RTT from a matched packet pair (Send, Echo) is defined as the difference between the timestamps of the Echo and the Send, respectively. From the packets collected shown in Figure 2, it is trivial to compute the RTT between the first packet: Send and its matched packet: Echo using their timestamps: RTT = 1,625,238,402.058117 − 1,625,238,402.018803 = 39,314 μs.

### 3.4. RTT and the Length of a Connnection Chain

The RTT from a matched packet pair (Send, Echo) in a connection chain is used to represent the length of a connection chain. We use the connection chain shown in Figure 1 to demonstrate the significance of a RTT and the rationale that the RTT from a matched packet pair can reflect the length of a connection chain. As we discussed, a Send is a request packet sent from Host 0, which is the first host of the connection chain and the packet can travel across all the connections along the chain and, finally, arrive at Host *N*, which is the last one of the chain. The total travel time for a Send from Host 0 to Host *N* contains the time consumed in each host and router between Host 0 and Host *N* and the propagation time in each connection of the chain. The time consumed in each host involves the packet processing time and the transmission time. This time duration dominates the value of RTT since the propagation time in each connection can be ignored. The more hosts connected in a connection chain, the larger the RTT value. Therefore, the RTT from a matched packet pair can tell us the number of hosts used in a connection chain.

Different hosts in the same connection chain may present a different processing time for the same Send packet. It is also true that the same host may use different processing time to handle different Send packets in the same connection chain. This makes the RTTs for different matched packet pairs fluctuated. In this research, other than using single RTT to represent the length of a connection chain, we use multiple ones: a collection of RTTs.

## 4. Rationale for Packet-Matching

Packet-matching technique is critical for detecting a stepping-stone intrusion. The detection approaches for a stepping-stone intrusion based on packet-matching cannot only significantly bring down the false positive detection errors, but also work effectively in resisting to attackers’ session manipulation.

Due to the complexity of computer network communication, matching TCP/IP packets is non-trivial but still possible. Some researchers, such as Paxson and Floyd [4], found a fact that packet RTTs follow the Poisson distribution approximately. This fact can be used to match TCP packets and further estimate the number of connections in an extended connection chain. In the following, we will discuss the rationale for packet-matching.

If we assume a variable *X* represent RTT values and its mean is µ and the standard deviation is δ, then the condition |X−µ|≤2δ must be satisfied if the RTTs following the Poisson distribution. This indicates that, for matched packet pairs, most of their RTTs are bounded around its mean µ within the distance of δ. This also tells us that the RTTs corresponding to a fixed length connection chain belong to the same cluster with µ as its center. Consequently, the RTTs of a connection chain in different lengths belong to different clusters. On the contrary, if we can obtain these clusters, it means that we can get the matched packet pairs.

If a connection chain with a fixed length is monitored, we sniff all the Send and Echo packets for a certain period. If we can get all the time differences between each Echo and all the Send packets before each Echo, we would obtain a data set in which some of the elements must be the RTTs for matched packets. Due to the Poisson distribution feature of RTTs, if the data set is clustered, the RTTs belonging to the same length of a connection chain must be classified into the same cluster. The pairs in the cluster with each pair containing a Send and an Echo are the matched packet ones for the Send and Echo packets sniffed.

There are many widely used data clustering methods, such as syntactic classifiers, minimum-distance classifiers, distance-function classifiers, statistical classifiers and MMD. With MMD, there is no prior ideas and knowledge about the final clustering results as well as the number of clusters. The Poisson distribution characteristic of RTTs from collected TCP packets motivates us to apply the MMD data mining algorithm to match network traffic.

## 5. Matching Packets through MMD Data Mining

In packet-matching, it is trivial to match an one-to-one Send and Echo pair. An one-to-one Send and Echo pair is a packet queue collected like {S_1_, E_1_, S_2_, E_2_, …, S*_n_*, E*_n_*} which contains *n* Send packets and *n* Echo packets. It is hard to match a Send with an Echo if the collected packet queue presents the format like {S_1_, S_2_, S_3_, E_1_, …, S*_n_*_−2_, S*_n_*_−1_, S*_n_*, E*_m_*}, or {S_1_, S_2_, …, S*_n_*, E_1_, E_2_, E_3_, …, E*_m_*}. In either of the cases, we do not know if E_1_ matches with either S_1_, S_2_, S_3_, …, or S*_n_*. The previous packet-matching algorithms, such as First-Match, Packet-matching Conservative algorithm and Packet-matching Greedy algorithm, cannot match them with a high matching rate. In the following, we first discuss the challenges to match TCP packets, then present our new approach MMD data mining algorithm to match Send and Echo packets regardless of the complexity of a collected packets queue.

### 5.1. Packet-Matching Challenges

The TCP/IP protocols was first developed by DARPA in the 1970s and used in ARPANET [24]. At 1970s, most computers were not efficient and network communication was in budding state. Almost all the designing efforts were dedicated to improving the efficiency of computing and communication. Computing and network security was not a focus at that time. Today, some security issues were caused by the inherent design of the TCP/IP protocols. Packet-matching is challenging because of some important designs to improve the efficiency of the TCP/IP protocols, other than communication security. Therefore, comprehending the working mechanism of the TCP/IP protocols is the key to understand the difficulty of packet-matching and the corresponding challenges.

In order to improve the efficiency of a computer network communication, the design of the TCP/IP protocols take the strategies of pipelining, cumulative acknowledgement and fast retransmission. The reliable data transfer in TCP/IP is implemented via packet acknowledgement, retransmission, error detection and loss detection. All of these techniques to improve the efficiency and reliability of a network communication make Send and Echo packets not a one-to-one match and bring challenges to packet-matching. This is the primary reason that most of the packet-matching algorithms suffer from low packet matching rate. In this section, we will discuss how the designing features of The TCP/IP protocols affect packet-matching.

TCP is a pipelined protocol. TCP packets from a sender can be sent out before they are acknowledged. A packet sent out, but not acknowledged must be buffered at the sender side. A buffer window size can be determined when a sender and a receiver establish a TCP connection. If a sender keeps sending without receiving an acknowledgement for the packets in fly, the buffer of the sender will be full eventually. The sender will stop sending if its buffer is full. If a packet in the buffer of a sender is acknowledged, it must be the oldest, unacknowledged packet in the buffer first. This makes the case of multiple consecutive Sends following by one Echo, such as, {S_1_, S_2_, S_3_, …S*_m_*, E_1_}. However, we are still not sure if E_1_ matches with S_1_ only, S_1_ and S_2_, or any other combinations of the Send packets. Since the sender and the receiver in a TCP communication are symmetric, it is possible that after the multiple Sends, we may collect multiple consecutive Echoes, such as the case of, {S_1_, S_2_, S_3_, …S*_m_*, E_1_, E_2_, …E*_n_*}. It is even harder to know which Send matches with which Echo(es). TCP supports a cumulative acknowledgement. This mechanism makes packet-matching even more difficult since one Echo may acknowledge multiple Sends.

When an OS command, such as in a Linux/Unix system, is keystroked by a user, each letter typed can incur a Send and maybe an Echo. At the end of the command, an “Enter” keystroke can make a Send and one or more Echoes since a Linux/Unix command executes at a receiver side may result in a large response returned back to the sender in multiple Echoes. In this situation, we can collect the packets as the following format: {S_1_, E_1_, E_2_, …, E*_m_*}.

If any error from a packet received by a receiver is detected, or packet loss is detected by a sender, a retransmission of the same packet will happen. The packet retransmission happens at the Transport layer. If a packet is detected lost, but actually not lost, the retransmission can make a duplication at the receiver side. Another case is that a packet is detected lost and actually lost cannot make any duplication at the receiver side. The packet retransmission can complicate the network communication, but does not affect packet-matching since all the packets collection is conducted at the application layer. Any retransmission or duplicated packets cannot be captured in this layer.

### 5.2. MMD Data Mining to Match TCP Packets

MMD (Maximum and Minimum Distance) data mining algorithm is used for massive data set classification. We propose to apply this algorithm to packet-matching since a packet queue can be converted to a data set.

Without loss of generality, we assume there is a queue of collected packets, such as {…S*_i_*, S*_i_*_+1_, …, S*_n_*, E*_j_*, E*_j_*_+1_, …, E*_m_*, …}. In this case, we do not know if E*_j_* matches with S*_i_*, or a group of Sends, such S*_i_*, S*_i_*_+1_, …, S*_i_*_+*k*_, where 1 < *k* < *n* − i; or if multiple Echoes match with a single S*_q_*, where *i* ≤ *q* ≤ *n*. If a connection is established and keeps its length unchanged while we collect the packets, as we discussed before, the RTT for any matched packet pair can represent the length of a connection chain. Different RTTs from different matched packet pairs may fluctuate, but their values are bounded or close because each of the RTTs is used to represent the length of the same connection chain. The data bounded in a large dataset can be mined into the same cluster via MMD.

If we have a packet queue {…S*_i_*, S*_i_*_+1_, …, S*_n_*, E*_j_*, E*_j_*_+1_, …, E*_m_*, …}, it is easy to compute the differences between each Send and all the Echoes after the Send. In a computer network communication, the possibility that a Send matches with an Echo that is far away from the Send is extremely small. If a request cannot be echoed in a certain time, the request will be resent and, thus, cause a lot of traffic. Therefore, each Send can match with an Echo that is not far away from the Send. We assume that each Send must be echoed within *w* Echoes after the Send; here, *w* is called a packet-matching window size. Under this assumption, a matched Echo for a Send can be found in *w* Echoes following the Send packet. Since we do not know which one of the *w* Echoes matches with the Send, we compute all the differences between the Send and all *w* Echoes after the Send. We repeat this for all the Sends in the queue with *n* Sends and *m* Echoes, a massive data set *X* = {*x*_1_, *x*_2_, …, *x_i_*, …, *x_q_*} with *q* = *n* × *w* timestamp gaps can be obtained. If we apply MMD data mining to the data set *X*, the cluster with n data elements (since it has *n* Send packets) is the one containing all the matched pair between each Send and its corresponding Echo. The following is the MMD data mining algorithm [25,26,27] we applied for packet-matching. In the algorithm, the inputs are Ts and Te, the timestamp data set for all the collected Sends and Echoes, respectively, a threshold t and packet-matching window size *w*. The dataset *X* is computed based on Ts, Te and *w*.
**Algorithm 1** Matching Send and Echo packets. **Input:**   **Ts = {ts*_i_*}:** timestamp data set for all the collected Sends   **Te = {te*_j_*}:** timestamp data set for all the collected Echoes   ***t*:** a threshold value which determines whether a new cluster should be created   ***w*:** packet-matching window size**Output:**   ***p*:** the number of cluster centers found;   ***C_j_*:** the cluster centers;   ***r_j_*:** the cluster sizes;   ***u_ij_*:** the indices of the original samples which belong to the *j*-th cluster;**MMD Algorithm (Ts, Te, *t*, *w*):****Begin**
0.Compute *X* = {*x_i_*} from Ts, Te, and *w*: *x_i_ = te_j_* − *ts_i_, …, x_i+w−_*_1_
*= te_j+w−_*_1_
*− ts_i_*1.$\mathrm{Set}\;C_{1}=x_{1},C_{2}=x_{j_{0}},u_{11}=1,u_{12}=j_{0}$$,\;\mathrm{where}\;\|x_{j_{0}}-C_{1}\|=\underset{2\leq i\leq L}{\max}\|x_{i}-C_{1}\|$Set *p* = 2,$\begin{array}{l} a=\bar{\left\|C_{i}-C_{j}\right\|}\;(arithmetic\;mean),\;\\ where\;1\leq i,j\leq p,\;i\neq j\;\\ and\;X^{\prime }=X-\left\{C_{1},C_{2}\right\}. \end{array}$2.Find *j*_0_, 1 ≤ *j*_0_$\;\leq \;p\;\mathrm{and}\;x_{i_{0}}\in X^{\prime }$ such that
$d=\left\|x_{i_{0}}-C_{j_{0}}\right\|=\underset{x_{i}\in X^{\prime }}{\max}\underset{1\leq j\leq p}{\min}\left\|x_{i}-C_{j}\right\|$If d<ta (no more clusters) go to step 4; otherwise go to step 3.3.Set $\begin{array}{l} p\leftarrow p+1,\;C_{\mathrm{Set}\;p+1}=x_{i_{0}},\;u_{p1}=i_{0}\\ X^{\prime }\leftarrow X^{\prime }-\left\{C_{p+1}\right\} \end{array}$, update *a*, and go to step 2.4.*r_j_* = 1, 1 ≤ *j* ≤ *p*5.$\mathrm{For}\;\mathrm{each}\;x_{i}\in X^{\prime }$find j:1≤j≤p$\;\mathrm{for}\;\mathrm{which}\;\left\|x_{i}-C_{j}\right\|=\underset{1\leq j\leq p}{\min}\left\|x_{i}-C_{j}\right\|$$\;\mathrm{and}\;\mathrm{set}\;r_{j}\leftarrow r_{j}+1\;and\;u_{r_{j}j}=i$, add *x_i_* to *C_j_*.6.For 1 ≤ *j* ≤ *p *repeat
$C_{j}=\frac{x_{u_{1j}}+x_{u_{2j}}+\ldots +x_{u_{r_{j}j}}}{r_{j}}$7.For 1 ≤ *j* ≤ *p* output *C_j_*.
**End**

In the Algorithm 1, as shown in Step 1, for the dataset *X* computed based on Ts, Te and *w*, it first determines all the cluster centers, then adds each element to the nearest cluster. It assumes that the first element *x*_1_ in *X* is the center of the first cluster C_1_, the center of the second cluster is the element $x_{j_{0}}$,which has the largest distance to C_1_. Compute the arithmetic mean α among all the existing cluster centers. The dataset is updated to be *X*′ by removing the two elements *x*_1_ and $x_{j_{0}}$. For each element in *X*, find the minimum distance to each existing cluster, then get the maximum one among all the minimum distances. This maximum distance is denoted as *d* in the algorithm. Compare *d* with the multiplication of *t* and α to determine if there are more clusters existed. If there exist more clusters, then repeat Step 2 and Step 3 as shown in the algorithm to find all the other clusters. Then, add each element left in *X* to each identified cluster in Step 5. Finally, compute the center for each cluster in Step 6. In order to help readers better understanding the MMD algorithm, we use the following flow chart (Figure 3) to represent the idea of the whole algorithm.

## 6. Experimental Results and Its Analysis

### 6.1. Experimental Setup

In our experiment, we initially established a connection chain with three hosts: the intruder’s machine, the sensor host and the first stepping-stone host S1. This chain is referred to as a 1-connection chain as it contains only one connection from the sensor to S1. Then, the chain is extended to the stepping-stone host S2 and referred to as a 2-connection chain that contains two connections from the sensor to S2. After that, the chain is extended to the stepping-stone host S3 and referred to as a 3-connection chain that contains three connections from the sensor to S3. Similarly, the chain is extended to the stepping-stone host S4 and referred to as a 4-connection chain and, finally, extended to the victim host and referred to as a 5-connection chain as there are five connections from the sensor to the victim host (see Figure 4).

In a LAN, the above connection chain of seven hosts in total was made by using seven local Ubuntu Linux systems in our department computer lab. In the Internet environment, we used two local Ubuntu Linux systems with one serving as an intruder host and the other one serving as a sensor. Then, we applied for five AWS Ubuntu virtual servers from Amazon AWS cloud. The servers are located in different parts of USA. The 1st AWS Ubuntu server with IP address 3.142.156.175, located in Ohio, USA, served as the first stepping-stone S1. The 2nd AWS Ubuntu server with IP address 54.161.133.239, located in North Virginia, USA, served as the second stepping-stone S2. The 3rd AWS Ubuntu server with IP address 13.57.216.213, located in California, USA, served as the third stepping-stone S3. The 4th AWS Ubuntu server with IP address 54.202.152.193, located in Oregon, USA, served as the fourth stepping-stone S4.

Finally, the 5th AWS Ubuntu server with IP address 3.15.144.130, located in Ohio, USA, served as the victim’s host. After all these AWS servers were ready, we used the tool OpenSSH to create a connection chain across the seven hosts as shown in Figure 4.

### 6.2. Data Collection

Data collection from a LAN: for the above 1-connection chain created in a LAN, we monitored and captured its TCP Send and Echo packets, matched them, then computed the RTTs of the capture packets from the matched packet pairs. The procedure was repeated 10 times and 10 packet Send and Echo datasets were collected for the 1-connection chain. Then, we repeated the similar tasks for the above 2-connection chain, 3-connection chain, 4-connection chain and 5-connection chain, respectively. Totally, we collected 50 collections of datasets from the LAN, each of which contains 10 datasets obtained from a downstream connection chain in the length of 1, 2, 3, 4, or 5, respectively.

Data collection from the Internet environment: We performed the similar tasks in the context of the Internet for the above five downstream connection chains in the length of 1, 2, 3, 4, or 5, respectively. Totally, we collected 50 collections of datasets from the Internet, each of which contains 10 datasets obtained from a downstream connection chain in different lengths.

#### 6.2.1. Collected Packets

We sniffed all the Send, Echo and Ack packets from the connection chain starting from the sensor to the end host and stored all the collected packets in a text file. In Figure 2, it shows the format of the first three packets captured. In each packet, we stored timestamp, source IP address, source port number, destination IP address, destination port number, TCP Flags, sequence number, acknowledgement number, windows size, options and the packet length. The timestamp in each packet is in the time unit of second. As shown in Figure 2, the first packet was captured at the time of 162,238,402.018803 s.

#### 6.2.2. Data Pre-Processing

Since the inputs of the MMD data mining packet-matching algorithm are timestamps for Send and Echo respectively, we processed the collected packets to obtain the timestamp in microsecond for each Send and stored them into the file Send.txt, as well as storing the timestamp for each Echo into the file Echo.txt. We call this step a data pre-processing stage. We made a Python code to complete the data pre-processing. In Figure 5, it shows the part of the contents from Send.txt and Echo.txt respectively.

#### 6.2.3. MMD Data Mining Output Demonstration

In the output of the MMD data mining packet-matching algorithm, it contains all the clusters identified. In each cluster, it shows the cluster center value and all the matched pairs line by line. Each line of matched pair involves the RTT value and the matched pair (*x*, *y*) where *x* represents the Send packet index number and *y* represents the Echo packet index number. All these indexes start from 0. Figure 5c shows the part of the output from a MMD data mining result. From this partial output, we can see three clusters. The first cluster has five matched packet pair and its center value is 39,767.839844 µs. The five matched pairs are (0, 0), (1, 1), (2, 2), (3, 3) and (4, 4) respectively. The pair (0, 0) indicates Send 0 matches with Echo 0. The second cluster has no matched pair, so its center value is -nan (ind). The third cluster has two matched pairs with cluster center value 64,697.0 µs. The cluster containing all or most of the Sends in its matched pairs is the one we are looking for because it represents the highest packet matching rate.

### 6.3. Packet-Matching Results and Its Analysis in a LAN

We first conducted an experiment in a LAN located in our department lab. The primary goal for this experiment is to verify the packet-matching correctness through MMD data mining. We identified 7 computers in the lab and impersonated Intruder’s host, Sensor host, S1, S2, S3, S4 and Victim’s host, respectively, as shown in Figure 4. We established a connection chain from Intruder’s host to Sensor host, then to S1 using OpenSSH, collected packets from Sensor host for ten times and stored the packets to ten different files respectively. Then, we extended the connection chain to S2, S3, S4 and Victim’s host to make a two, three, four and five-connection chain respectively and repeated the packets collection process in Sensor host. We processed the collected packets to obtain ten Echo.txt and ten Send.txt files under the case that the connection chain had one, two, three, four, or five connections respectively. In each time of the data collection, we controlled the keystroke at Intruder’s host to make all the collected packets to be one-to-one match that was in the format of {S_1_, E_1_, S_2_, E_2_, …, S*_n_*, E*_n_*}. It is easy to match the collected packets manually and compared with the packet-matching results via MMD data mining to verify the accuracy of MMD packet-matching algorithm. We typed 25 times and sniffed 25 Sends and 25 Echoes. The keystrokes were repeated ten times under the chain of one-connection, two-connection, three-connection, four-connection and five-connection, respectively. The packet-matching results from MMD algorithm have shown that all the Sends and Echoes can be matched 100% correctly. Table 1 shows the average packet-matching rate from MMD data mining can reach 100%.

### 6.4. Packet-Matching Results and Its Analysis in the Internet

We conducted the same experiment with one to five connections set up in the Internet and collected the packets at Sensor host. However, the hosts S1, S2, S3, S4 and Victim’s host are AWS servers mentioned in the above. The results in Table 2, Table 3 and Table 4 show the ten times packet collection and matching results for one, three and five connections of the connection chain respectively. In each time of the data collection, each table shows the number of Send collected, the number of Echo collected and the number of Send matched. The matching rate is the ratio between the number of Send matched with the number of Send collected.

The results also show that under the Internet context, the number of Send and Echo packets collected cannot be in one-to-one case regardless the length of a connection chain. The number of Echo packets is more than the number of Sends partially because some Echoes may be from the execution results of a Linux command. That is why we calculate the matching rate based on the number of Sends collected, other than the Echoes. The average packet-matching rate from MMD data mining is 94.35% for one-connection chain, 94.97% for three-connection chain and 95.88% for five-connection chain.

### 6.5. Comparison with the Existing Packet-Matching Approaches

The packets in a computer network communication can be matched using packet sequence number, Yung’s approach [10], Step-function [2], Packet-matching conservative algorithm [3] or Packet-matching greedy algorithm [3]. In Table 5, we summarize all the packet-matching algorithms and list their matching mechanism, advantage and disadvantage, respectively.

We set up an experiment to verify the matching rate between all the existing algorithms and MMD algorithm. In this experiment, we set up a connection chain shown as in Figure 4. In this chain, the Intruder’s host and the Sensor host were located in our lab, all other stepping-stone hosts and the Victim’s host were the AWS servers we identified. We collected 35 to 38 Send packets and all the echo packets incurred by the Send packets from the outgoing connection of the Sensor host. We run the above different algorithms to match the Send and Echo packets collected and compare with the packet-matching result obtained from MMD algorithm.

In Table 6, it shows the comparison results in terms of packet-matching rate, number of Send packets collected, number of Echo packets collected and the number of Send packets matched. The packet-matching rate is the ratio between the number of Send matched and the number of Send collected. From the comparison result in the table, we can see that MMD data mining algorithm outperforms all the other algorithms in terms of matching rate.

## 7. Conclusions and Future Work

Estimating the length of an interactive TCP connection chain used by attackers to launch their attacks has become a primary way to detect stepping-stone intrusions. We found that the length of a TCP connection chain can be approximated by the RTTs between the Sends and Echoes sniffed from the chain. In order to compute the RTTs using captured Sends and Echoes, the Sends and Echoes must be matched, otherwise, a wrong RTT would be obtained. A matched Send and Echo packet pair can be used to compute the RTT between the Echo and the Send.

In this paper, we first define a Send and an Echo packet from TCP packets and present the idea to match a Send with an Echo. Challenges of matching Sends and Echoes under TCP protocol are also discussed. We second model a network traffic as the collection of RTTs from Sends and Echoes. The point is how to match Sends and Echoes collected to obtain the correct RTTs.

Packet-matching has been widely studied and applied to a stepping-stone intrusion detection. There have been a couple of algorithms developed to match Send and Echo packets. Most of them suffer from a low matching rate. The state-of-the-art packet-matching algorithm, packet-matching greedy algorithm, can only reach around 63% packet matching rate. We applied the MMD data mining algorithm to match TCP Send and Echo packets and found that it outperforms all the existing packet-matching algorithms including state-of-the art packet-matching greedy algorithm. The experimental results from a local area network shows the packet-matching accuracy obtained from the MMD data mining algorithm is 100%. We also found from the experimental results conducted under the Internet context that the packet-matching rate from MMD can reach around 94% on average.

As we mentioned before, the algorithms to detect stepping-stone intrusion are classified into two different categories: HSSID and NSSID. Most of the approaches to detect stepping-stone intrusion in HSSID category does not need to match network traffic, so MMD data mining algorithm is not the best for HSSID. However, it is applicable to NSSID since most of the algorithm developed to detect stepping-stone intrusion in the category of NSSID need packet matching.

In the future, there are a few things deserved to explore. We found that if applying MMD to match thousands and even more Sends and Echoes, our C code program needs to run relatively a long time to get the final packet-matching result. This indicates MMD is not efficient in term of time complexity. The first future research work is to improve MMD algorithm in packet-matching to reduce its time complexity. In order to counter stepping-stone intrusion detection, most attackers tend to manipulate the connection chains employed to launch their attacks. Chaff-perturbation evasion is a typical manipulation exploited by attackers to evade stepping-stone intrusion detection. The second future work would be to test the resistance performance to chaff-perturbation manipulation for the MMD data mining packet-matching algorithm. That is to study, when injecting meaningless TCP packets into a connection chain with the chaff-rate of 10%, 20%, 30%, … and even more, how the packet-matching rate from MMD is affected. Time-jittering is another popular way employed by intruders to evade stepping-stone intrusion detection. In time-jittering evasion, the Send and Echo packets in a connection chain can be delayed randomly by intruders to evade detection. The third future work is to explore the resistance performance of the MMD data mining algorithm to time-jittering manipulation with 10%, 20%, 30%, … and even more percentage of the packets are randomly delayed.

## Figures and Tables

**Figure 1 sensors-21-07464-f001:**
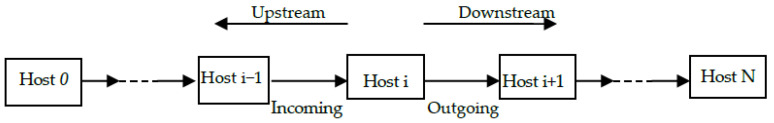
A Sample Connection Chain.

**Figure 2 sensors-21-07464-f002:**
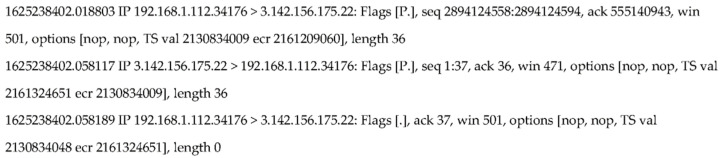
Three TCP packets collected.

**Figure 3 sensors-21-07464-f003:**
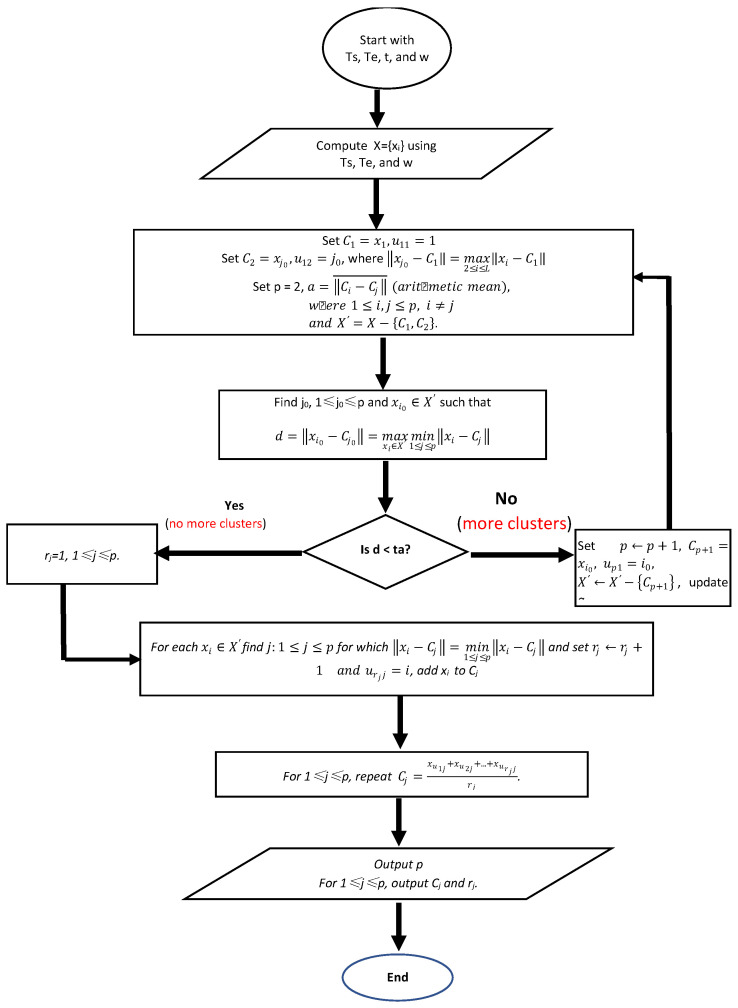
The flow chart for the Packet matching Algorithm.

**Figure 4 sensors-21-07464-f004:**
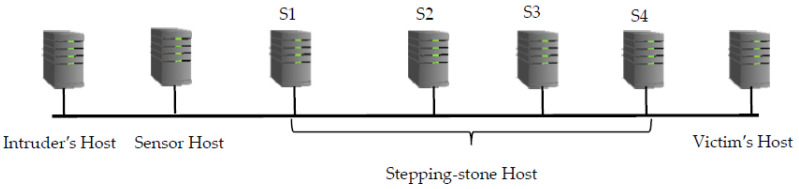
A connection chain of length five from the sensor host to the victim’s host.

**Figure 5 sensors-21-07464-f005:**
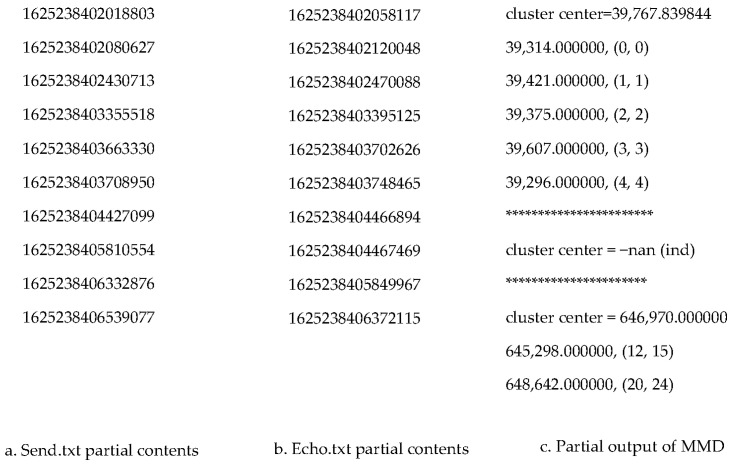
Input file format and output from MMD Data Mining Packet-matching.

**Table 1 sensors-21-07464-t001:** Average packet-matching rate of MMD data mining.

	Match Rate and Packet Number	Packet-Matching Rate (Average from Ten Times)	Number of Send Collected	Number of Echo Collected	Number of Send Matched
Connection Chain Length					
One connection		100%	25	25	25
Two connections		100%	25	25	25
Three connections		100%	25	25	25
Four connections		100%	25	25	25
Five connections		100%	25	25	25

**Table 2 sensors-21-07464-t002:** Packet-matching rate in one-connection (average rate: 94.35%).

	Match Rate and Packet Number	Packet-Matching Rate (%)	Number of Send Collected	Number of Echo Collected	Number of Send Matched
Times to Repeat Data Collection					
1st time		93.94	33	37	31
2nd time		96.88	32	40	31
3rd time		96.67	30	34	29
4th time		96.77	31	35	30
5th time		83.87	31	34	26
6th time		96.97	33	37	32
7th time		100.00	33	37	33
8th time		90.91	33	39	30
9th time		87.50	32	39	28
10th time		100.00	36	43	36

**Table 3 sensors-21-07464-t003:** Packet-matching rate in three-connection (average rate: 94.97%).

	Match Rate and Packet Number	Packet-Matching Rate (%)	Number of Send Collected	Number of Echo Collected	Number of Send Matched
Times to Repeat Data Collection					
1st time		100.00	35	45	35
2nd time		97.14	35	48	34
3rd time		85.71	35	41	30
4th time		100.00	38	50	38
5th time		97.14	35	42	34
6th time		94.29	35	45	33
7th time		94.29	35	46	33
8th time		94.29	35	44	33
9th time		97.14	35	43	34
10th time		89.74	39	49	35

**Table 4 sensors-21-07464-t004:** Packet-matching rate in Five-Connection (average rate: 95.88%).

	Match Rate and Packet Number	Packet-Matching Rate (%)	Number of Send Collected	Number of Echo Collected	Number of Send Matched
Times to Repeat Data Collection					
1st time		91.43	35	41	32
2nd time		91.43	35	41	32
3rd time		94.29	35	41	33
4th time		97.14	35	41	34
5th time		92.11	38	44	35
6th time		100.00	35	41	35
7th time		100.00	35	40	35
8th time		95.24	42	46	40
9th time		100.00	35	41	35
10th time		97.14	35	42	34

**Table 5 sensors-21-07464-t005:** Comparison among different packet-matching algorithms.

	Items to Compare	Packet-Matching Mechanism	Advantage	Disadvantage
Algorithms				
Sequence number-based		Using packet sequence numbers	Easy to implement	Few packets matched in the Internet
Yung’s approach		Each Send always matches with the closest Ack, or Echo packet	Easy to implement	May introduce wrong matchings
Step-function		Each Echo always matches with the first Send before the Echo packet	Efficient and each to implement	Low packet matching rate
Packet-matching conservative algorithm		The packets meet the following condition can be matched: Send.Ack = Echo.Seq (1a) Send.Seq < Echo.Ack (1b)	High rate of packet matching accuracy	Low rate of packet matching
Packet-matching greedy algorithm		Matched packets meet the following conditions: Send.Ack ≤ Echo.Seq (1a) Send.Seq < Echo.Ack (1b)	May introduce false-positive errors for packet matching	It can get higher packet matching rate than the above algorithms
MMD data mining algorithm		Makes use of the distribution of RTTs—Poisson Distribution	Highest rate of packet matching accuracy	Highest packet-matching rate

**Table 6 sensors-21-07464-t006:** Packet-Matching algorithm performance comparison under the Internet.

	Matching Rate	Packet-Matching Rate (%)	Number of Send Collected	Number of Echo Collected	Number of Send Matched
Matching Algorithms					
Sequence Number		14.28	35	45	5
Yung’s approach		28.57	35	42	10
Step-function		22.86	35	40	8
Packet-matching conservative algorithm		37.14	35	42	13
Packet-matching greedy algorithm		62.85	38	44	22
MMD data mining		94.44	36	42	34

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
