# Peer review of "Applying MMD Data Mining to Match Network Traffic for Stepping-Stone Intrusion Detection"

_sensors, 2021, doi:10.3390/s21227464_

Round 1

Reviewer 1 Report

  1. Abstract need revision to be in line with conclusion
    2. Conclusion need to be revised to be in line with the research findings
    3. A thorough proof-read is required
    4. A few of the references are not in complete form, please check all the references carefully

Number of references are not enough and may be increased. further, Some related references are missing and can be added such as

Wang, Lixin, Jianhua Yang, Michael Workman, and Pengjun Wan. "Effective algorithms to detect stepping-stone intrusion by removing outliers of packet RTTs." Tsinghua Science and Technology 27, no. 2 (2021): 432-442.

Almufareh, F. IoT Wireless Intrusion Detection and Network Traffic Analysis.

Author Response

see the uploaded file. 

Reviewer 2 Report

The paper title “Applying MMD Data Mining to Match Network Traffic for Stepping-stone Intrusion Detection” addressed a very interesting problem of detecting the Stepping-stone Intrusion in the network. However, the following comments need to be addressed in order to increase the readability of the paper.

  1.     In the introduction, the author included section 1.5 as Contribution and Novelty, however the section lack to express the complete novelty of the work.
  2. There are a few grammar issues. Please double-check the draft paper.
  3. Section 3 of the manuscript is about “Modeling Network Traffic” which require a more in-depth explanation. This section does not fully cover the explanation of using the existing tools, e.g s/w ,h/w etc.
  4. Section 5.2 covers the sudo code of the proposed MMD algorithms, however, the algorithms will be more clearly visible if a graphical flow chart is added.
  5. A concrete conclusion is needed at the end of the manuscript,  whether the proposed MMD algorithms is best for Host-based Stepping-Stone Intrusion Detection (HSSID) or Network-based Stepping-Stone Intrusion Detection (NSSID). Please discuss this in conclusion (if possible).
  6. Also, add few more future directions. As 'exploring the performance of MMD' is not enough.

Author Response

see the uploaded file. 

Reviewer 3 Report

The manuscript proposes a method to detect stepping stone intrusion using MMD approach. They claim to achieve the matching rate of around 94% on an average. The following points are observed.

  1. In abstract the author claims that the proposed approach outperforms the existing packet matching algorithms. However, it is better to provide the parameters of comparison to support the claim.
  2. Section 1, has no content and directly section 1.1 has been started. It is suggested to give brief description of the flow of the manuscript in section 1 before going to any subsections.
  3. The complete work is focused on stepping stone intrusion detection; however, a very brief description is provided in section 1.1. It is suggested to provide a more detailed description about it and it should be supported by relevant references.
  4. Section 2 (literature Review), the starting sentence “A great number of detection approaches for stepping-stone intrusion…” The language is not appropriately used here. It is suggested to revise the sentence with some relevant contextual word selection.
  5. In the literature review section, it is merely discussing what is being done by the other authors without any analysis of the research gaps or identifying the existing limitations. It is suggested to provide a tabular comparison of the related works and appropriately identify the research gaps.
  6. Figure 3 is not clearly visible. It is suggested to use a better quality image.
  7. All the references, table and figures must be cited in the body of the text.
  8. Although a lot of research is going in this particular field, however only one reference of the year 2021 is being used in the manuscript. It is suggested to explore few more latest state-of-the-art research works in this domain.  

Round 2

Reviewer 3 Report

The suggested revisions have been incorporated in the updated manuscript.